# Transcriptomic Analysis of Respiratory Tissue and Cell Line Models to Examine Glycosylation Machinery during SARS-CoV-2 Infection

**DOI:** 10.3390/v13010082

**Published:** 2021-01-08

**Authors:** Anup Oommen, Stephen Cunningham, Lokesh Joshi

**Affiliations:** 1Advanced Glycoscience Research Cluster (AGRC), National University of Ireland Galway, H91 TK33 Galway, Ireland; anupmammen.oommen@nuigalway.ie (A.O.); stephen.cunningham@nuigalway.ie (S.C.); 2Centre for Research in Medical Devices (CURAM), National University of Ireland Galway, H91 TK33 Galway, Ireland

**Keywords:** glycosylation, transcription, immune function, infection, susceptibility, biomarker, therapeutic

## Abstract

Glycosylation, being the most abundant post-translational modification, plays a profound role affecting expression, localization and function of proteins and macromolecules in immune response to infection. Presented are the findings of a transcriptomic analysis performed using high-throughput functional genomics data from public repository to examine the altered transcription of the human glycosylation machinery in response to SARS-CoV-2 stimulus and infection. In addition to the conventional in silico functional enrichment analysis methods we also present results from the manual analysis of biomedical literature databases to bring about the biological significance of glycans and glycan-binding proteins in modulating the host immune response during SARS-CoV-2 infection. Our analysis revealed key immunomodulatory lectins, proteoglycans and glycan epitopes implicated in exerting both negative and positive downstream inflammatory signaling pathways, in addition to its vital role as adhesion receptors for SARS-CoV-2 pathogen. A hypothetical correlation of the differentially expressed human glycogenes with the altered host inflammatory response and the cytokine storm-generated in response to SARS-CoV-2 pathogen is proposed. These markers can provide novel insights into the diverse roles and functioning of glycosylation pathways modulated by SARS-CoV-2, provide avenues of stratification, treatment, and targeted approaches for COVID-19 immunity and other viral infectious agents.

## 1. Introduction

Of the four major classes of biomolecules (nucleic acids, proteins, glycans, and lipids), the glycans conjugated to proteins and lipids provide a ubiquitous biological interface at cell surfaces, where they are instrumental in intermolecular and cell–cell recognition events that define and control cell interactions and functions [1]. Indeed, with greater than 70% of circulating proteins being glycosylated, including most cytokines and all immunoglobulins, the role of glycosylation in immune function is key, extending beyond the first-contact cell surface interaction [2]. Though not template-driven, the synthesis and presence of glycans on cell surfaces and macromolecules is driven by a known and diverse set of genes, collectively referred to as ‘glycogenes’, which encode the glycosylation machinery. Glycan recognition and function is typically mediated by complementary glycan-binding proteins (GBPs), which present specific glycan recognition domains conferring glycan-binding specificity. In addition, functional domains translate glycan recognition into functional cellular responses, positioning the role of glycans and GBPs as key components of the immune system [3]. Greater understanding of the glycans, GBPs and their associated molecular machinery, which underlie and regulate immune function, will permit a greater biological understanding of host and coronavirus SARS-CoV-2 interaction. Presenting opportunities for rational design of glycan and GBPs targeted therapeutics, immune dysfunction measured via glycan modulation, and potential risk and stratification of individuals based on glycan signatures, through the utilization of glycan biomarker diagnostics.

As the significance of SARS-CoV-2 and the associated risks posed more evident, diverse structural, biochemical and genome-wide screening approaches have been undertaken to identify and elucidate host factors involved in SARS-CoV-2 infection [4,5]. Studies have highlighted the relevance played by glycosylation on host cell receptors and viral membrane envelope proteins in facilitating host–virus infection [6,7,8,9]. Findings to date have indicated host-virus interaction involvement with sialic acid biosynthesis and sialylation pathways, as well as in the regulation of cell-intrinsic immunity to be critical for supporting different stages of viral replication, including entry and antiviral responses. This is in keeping with reported glycosylation involvement in human immunodeficiency virus-1 (HIV-1), hemagglutinin glycoprotein (HA) of influenza virus, coronavirus glycoprotein spike (S) SARs, glycoprotein (GP) of Ebola virus, glycoprotein complex (GPC) of Lassa virus, and envelope (E) glycoprotein of dengue, Zika, and other flaviviruses [10,11,12,13]. Beyond host-virus infection, a study by Sadat et al. examining a rare genetic disease (type II congenital disorders of glycosylation (CDG-IIb)), caused by mutation in the gene encoding N-linked glycan processing enzyme mannosyl-oligosaccharide glucosidase (MOGS), reported that despite severe hypogammaglobulinemia, the patients did not show susceptibility to viral infection or recurrent infections, indicating a potential role of glyco-machinery in directly impacting an individual’s degree of susceptibility to enveloped viruses [14]. Establishing an indispensable and critical role of glycosylation machinery in viral replication, with altered glycosylation of host and viral proteins conferring potential resistance to virus infection.

Modulation of host glycosylation machinery is a fundamental molecular mechanism leveraged by biological agents for eliciting pathogen specific immunity [15]. Variation in glycosylation patterns of host proteins [16,17] especially the antigen specific antibodies [18], have been extensively characterized for viral agents such as the H1N1 influenza virus, hepatitis B virus and HIV-1 to determine the relevance of glycosylation in regulating both innate and adaptive immune response. In the current COVID-19 pandemic, successful infection of host airway epithelial cells by the heavily glycosylated SARS-CoV-2 virus is dependent upon the occupancy and nature of glycan chains expression on the virus binding sites of human Angiotensin Converting Enzyme-2 (ACE2) protein receptors [7]. Interestingly, a recent study using high throughput pseudovirus-based neutralization assay identified 80 natural variants and 26 glycosylation spike mutants of SARS-CoV-2 which significantly affected the viral infectivity [19]. Moreover, glycosylation is a major modulatory factor which influences the nature of inflammatory responses [20,21], with altered expression of glycan chain structures reported on several proteins and tissues underlying many acute and chronic respiratory diseases [22,23,24,25,26,27]. Interestingly, these diseases share pulmonary pathologies similar to observations reported in COVID-19 subjects [28]. These observations and supporting evidences highlight the need and potential to evaluate whether SARS-CoV-2 infection induces transcriptional alteration of host glycosylation machinery in order to facilitate different stages of viral replication as well as to modulate host immune response. At a preliminary stage of the hypothesis validation, we utilized high throughput gene expression data from the Gene Expression Omnibus database repository, generated from human subjects and cell lines infected by SARS-CoV-2 virus.

## 2. Materials and Methods

### 2.1. Gene Expression Data Selection

Gene Expression Omnibus (GEO) was queried using the search terms “(COVID-19 OR SARS-COV-2) AND gse[entry type]”, with filtering criteria “Expression profiling by HTS” and the organism “Homo sapiens” applied. Datasets included gene expression data generated from ex-vivo and autopsy samples from COVID-19 subjects as well as human cell lines (*n* ≥ 3) and organoids treated with SARS-CoV-2 virus (*n* ≥ 3). Datasets without gene annotation were excluded.

### 2.2. Glycosylation Process Related Gene Set

Glycosylation machinery gene set (Glycogenes—metabolic genes, transporters and transferases) was compiled from the GlycoGAIT database [29]. Summarily, data from Kyoto Encyclopedia of Genes and Genomes (KEGG), ExplorEnz—The Enzyme Database, GlycoGene database (GGdb), Consortium for Functional Glycomics (CFG), UniProt and from the textbooks—Essentials of Glycobiology and Handbook of Glycomics was extracted using keywords centred around different sugar moieties involved in glycosylation. Uniform nomenclature was maintained using HUGO Gene Nomenclature Committee (HGNC) database as a reference. Using proteoglycan and lectin as keywords, information for glycan binding proteins and proteoglycans were also extracted from the HGNC database (https://www.genenames.org/) and further cross-validating the list using the Gene group reports from HGNC for completion. Details of the enzymatic reactions for the glycosyltransferase and glycosidase enzymes was enriched by manually curating the reactions from the BRENDA enzyme database (https://www.brenda-enzymes.org/index.php) and ExPASy bioinformatics resource portal (https://www.expasy.org/). For interactions where the reaction information is not available the interactions were curated manually from PubMed sources.

### 2.3. Data Processing, Functional Enrichment Analysis and Network Visualization

GenePattern (http://software.broadinstitute.org/cancer/software/genepattern/) [30] and Galaxy (https://usegalaxy.org/) [31] were utilized for data processing as detailed in their respective user manuals. Where datasets had existing processed result files available through the GEO database these were used. Hierarchical clustering of the normalized gene expression data was performed using the Heatmap w ggplot tool in Galaxy Version 2.2.1. The mapping of differentially expressed glycogenes (DEGs) to known signaling pathways and cellular processes, and gene set enrichment analysis (GSEA), were performed using the g:Profiler web server [32]. Using the “Retrieve/ID mapping tool” [33] available in the UniProt Knowledgebase [34] detailed gene/protein function and other related database reference IDs were extracted for the DEGs. Pathway analysis was performed using the Reactome biological pathways (https://reactome.org/) [35]. The induced network module function available in ConsensusPathDB—(CPDB—http://cpdb.molgen.mpg.de/) [36] was used to identify any possible functional relationship between DEGs coding for lectins through the protein–protein interaction and biochemical reactions. Network analysis and visualization was performed with Cytoscape software (http://www.cytoscape.org/) [37].

## 3. Results

Seven array datasets were identified from the GEO database (July 2020), with the filter criteria of having at least three minimum samples for both control and SARS-CoV-2 treated/infected conditions (Appendix A). From these datasets, data from biological samples and cell lines, relevant only to the upper respiratory tract infection, were selected yielding six data points for subsequent data processing using the DESeq2 algorithm available in Genepattern genomics tool (Appendix A). From the DESeq2 analysis results (normalized, log2 fold changed, Benjamini–Hochberg adjusted *p*-value) significant genes with adjusted *p* value ≤ 0.1 and uncorrected *p* value ≤ 0.05 were identified for each data points (Appendix A). Using the gene set from the GlycoGAIT database (Appendix A), DEGs for each data points were identified which constitute ~3% of the total differentially expressed genes under the selected *p* value cut off, except for the lung samples (Appendix A). Data analysis of each data points using the frequency distribution function in Excel revealed that distribution of differentially expressed genes from the SARS-CoV-2 infected cell lines and organoids are largely represented within the range of −0.5 to +0.5 log twofold change values. On the contrary for the biopsy samples (nasopharyngeal and lung) the gene distribution was higher above the ±0.5 log twofold change cut-off values (Appendix A). Moreover, the clustered heatmap of the DEGs revealed that the gene expression pattern obtained from the nasopharyngeal swab and the Calu cell line show close similarity and were also the two data sources that showed maximum number of significant differentially expressed genes including the glycogenes (Appendix A). Higher level gene family association of the DEGs revealed that glycosyltransferases, lectins, proteoglycans, glycosidases and sulfotransferases as the maximum represented groups, majorly in the Calu cell line and the nasopharyngeal samples (Figure 1A,B, Appendix A).

For focused analysis on glycogenes, a stringent log2 fold change cut-off values ≤ −0.58 and ≥0.58 (fold change ≥ 1.5-fold) was performed. Detailed gene set enrichment analysis of the resulting DEGs using the g:Profiler toolset, highlighted specific cellular processes such as the glycosaminoglycan, proteoglycan, glycolipid, N-glycan and O-glycan metabolic processes as the maximum represented biological categories (Appendix A). Similarly, acetylglucosaminyltransferases, acetylgalactosaminyltransferases, carbohydrate binding, sulfotransferases were the most significant and more specific GO: molecular functions identified from the g:Profiler toolset (Figure 2A,B); Appendix A). Combined analysis of the gene family association and gene enrichment analysis results revealed that the maximum represented glycosyltransferase gene family were associated with the glycolipid, N-glycan and O-glycan metabolic processes, which also included the fucosyltransferases and sialyltransferases involved in the synthesis of diverse glycan epitopes.

A genome-wide overview analysis from the Reactome database highlighted metabolic pathways associated with glycosylation modifications and additionally a number of immune related pathways including the immune-regulatory interaction between lymphoid and nonlymphoid cells, majorly represented by the DEGs transcribing for the lectin proteins (Appendix A). Among the 56 metabolic processes that were mapped to the Reactome database, pathways with maximum number of intersecting DEGs were those associated with N-linked glycosylation (precursor biosynthesis, antennae elongation, trimming and complex type N-glycan synthesis); glycosaminoglycan metabolism (chondroitin/keratan/dermatan/heparin sulfate metabolism); O-linked glycosylation (biosynthesis and termination) and glycosylphosphatidylinositol (GPI) anchor biosynthesis. The DEGs associated with fructosyltransferase genes where mapped majorly to the blood group and Lewis antigen biosynthesis pathways (Appendix A). A representative pathway image of the cellular process from the Reactome database—“Blood group system biosynthesis”, highlighted using the gene expression data obtained from the human nasopharyngeal swab of Covid-19 patients is given in Appendix A. 

Lectins were the second category of maximum represented DEGs identified from the SARS-CoV-2 infected biopsy samples, cell lines and organoid relevant to the respiratory tract tissues (Appendix A). The C-type lectin receptors (Group II and Group V-NK cell receptor categories—CLEC) and the sialic acid binding I-type lectin receptors (SIGLEC) were the predominant ones with upregulated gene expression pattern (Appendix A). In order to delineate more detailed functional significance of the differentially regulated lectins, we used the induced network module feature from the CPDB database [36]. The protein–protein interaction network revealed molecular network association of lectin DEGs with both the negative regulatory and immunostimulatory signaling pathways. For example, the network captures the interaction between the galactose binding lectins (LGALS 1/2/3) with the negative regulatory protein tyrosine phosphatase (PTPN) signaling pathway while the CLECs and SIGLECs were found to be associated with the immunostimulatory signaling pathways mediated by the toll like receptors (TLRs), Killer Cell Lectin Like Receptor (KLRs), MHC class I-related molecules and the intracellular non-receptor tyrosine kinase signaling pathways (Figure 3). Though the role of such interactions needs to be experimental assessed in a SARS-CoV-2 infected airway epithelium system, the analysis network findings highlight the prospective immunomodulatory role of these lectins, which are well-characterized positive/negative regulators of innate and adaptive immunity [38].

Significant DEGs, grouped under the glycosylation associated cellular function pathways, were also manually analyzed for the association with human pathogenic viruses and immune system in order to understand the physiological relevance in response to SARS-CoV-2 infection from the biological samples (biopsies and cell lines) representing the upper respiratory tract (Appendix A). Manual compilation of the glycan epitope association for the DEGs, enabled the identification of the clinical significance of the transcriptomic changes for few genes in relation with immune response and viral infection conditions (Appendix A). For example, serglycin (SRGN) (increased expression pattern across samples), is one of the main proteoglycans of the cytotoxic granules in CTLs and NK cells and is known to regulate the kinetics of antiviral CD8^+^ T-cell responses. Moreover, serglycin proteoglycan with sulfated glycosaminoglycans of either heparin, heparan sulfate or chondroitin sulfate types attached to it is a major regulator of the mast cell secretory granule homeostasis [39]. Recent studies reported the presence of activated mast cells in the lungs of deceased patients with COVID-19, which are correlated with the release of pathogenic mediators linked to pulmonary edema, inflammation and thromboses [40]. Given the heparan sulfate-dependent enhanced attachment and infection of SARS-CoV-2 virus [41], it will be interesting to evaluate a model where serglycin as a critical mediator between SARS-CoV-2 infection, mast cell activation and the pulmonary pathology.

Similarly, increased expression of major genes involved in Core-2 O-glycan synthesis correlate with a recent report regarding the SARS-CoV-2 spike RBD protein glycosylation by host cell lines. The increased gene expression pattern of β-1,4 N-acetylgalactosaminyltransferase 2 (B4GALNT2), involved in the synthesis of Sd(a): (Neu5Acα2,3-[GalNAc-β1,4]Gal-β1,4-GlcNAc) antigen and the fucosyltransferases as well as α-2,3-sialyltransferases involved in the synthesis of Sialyl Lewis x/a (sLe^x/a^): (Neu5Acα2,3-Galβ1-4-[Fucα1-3]-GlcNAcβ) antigens, may suggest the intriguing possibility of expression of these antigens in the upper and lower tract of airway epithelium. The role of Sd(a) and sLe^x/a^ antigens as glycan ligands in regulating the adhesion and infection of airway epithelium by human and avian influenza viral strains is well reported (Appendix A). These antigens also plays a crucial role in regulating the lytic function of cytotoxic T lymphocytes as well as leukocyte adhesion and migration and hence might also correlate with the SAR-CoV-2 infection and immune overdrive (Appendix A).

Upregulated expression of genes encoding lectins which are previously reported to function as adhesive receptors for SARS-CoV-2 (such as SIGLEC5, MASP2, CLEC4M) were identified. Among them the mannose binding lectin serine protease 2 (MASP-2) has also been implicated in the altered lectin complement pathway resulting in accelerated inflammatory responses and lung damages involved in COVID-19 pathogenesis [42]. Few other C-type, I-type and S-type lectins in the list were found to be either induced by or associated with the adhesion of other viral strains such as the influenza virus, Ebola virus, Hepatitis C virus, Human Immunodeficiency Virus. Among Galectins, increased expression of LGALS9 elicited by SARS-CoV-2 infection correlate with that observed during the HIV-1 infection which was reported to mediate T-cell inflammation and exhaustion [43] as is the case with COVID-19. From an immune regulatory perspective, previous reports implicate both positive and negative regulatory functions for various DEGs belonging to the lectin family. The regulatory functions broadly include negatively regulating antiviral signaling, NK cell cytotoxicity, coagulation cascade and T-cell exhaustion while the positive regulatory functions include activation of the pro-inflammatory responses from lymphocytes, neutrophils and monocytes as well as immune cell infiltration, cell adhesion, motility and migration (Appendix A). 

The downregulated DEGs primarily represent the biological processes involved in glycan precursor biosynthetic pathways such as the N-glycan precursor biosynthetic processes, O-linked biosynthetic process, GPI-anchor biosynthesis as well as fucose, mannose and galactose metabolism (Appendix A). Defective GPI-anchor biosynthetic pathway as well as the N-glycan chain transfer to the nascent polypeptide chain were shown previously to affect the infectivity of pathogenic viruses like HIV-1 and Hepatitis C virus [40,41]. Whether this downregulated expression pattern represent an adaptive feedback mechanism can only be speculated and can be subjected to targeted experimental validation to explore the regulatory mechanism under SARS-CoV-2 infection condition.

## 4. Discussion

Similar to the strategies previously applied for curbing the spread of viral infections, current research efforts are directed towards the development of therapeutic antibodies and vaccines for effective COVID-19 treatment [44,45]. Kinetics of immune response from mild, moderate to severe COVID-19 patients ranges from augmented humoral and cellular immunity to lymphopenia affecting CD4+ T cells, CD8+ T cells, B cells and natural killer cells [46,47,48,49]. Furthermore, the degree of lymphopenia and pro-inflammatory cytokine storm was reported to be higher in severe COVID-19 patients and correlated with the adverse outcomes such as immune mediated lung injury and acute respiratory distress syndrome in these patients [50,51,52,53]. Reduction in the level of ACE2, following the binding and internalization of SARS coronavirus is considered as a critical factor that alters Renin-angiotensin system and concomitant disease pathogenesis such as lung oedemas, elevated macrophage infiltration, cytokine production and endothelial dysfunction [54,55]. The exponential growth in the cytokine storm in lung tissues and systemic circulation beget the pulmonary pathology of SARS-CoV-2 infection alveolar damage with cellular fibromyxoid exudates, desquamation of pneumocytes and hyaline membrane formation which are the salient features of ARDS [52,56]. A recent review on the altered expression of different cytokines in COVID-19 patients concluded that SARS-CoV-2 infection could be characterized by the depletion of antiviral defenses (such as delayed IFN-α and -β response as well as an elevated production of inflammatory cytokines [57]. Further to, altered procoagulant–anticoagulant pathways associated with the cytokine storm also drives the development of microthrombosis, disseminated intravascular coagulation, and multi-organ failure as evidenced in severe cases of COVID-19 subjects [58,59]. Despite extensive characterization of the altered immunological features underlying the spectrum of disease associated with COVID-19, the role played by alteration in glycosylation PTM remains to be addressed.

Glycan binding proteins (GBPs) and the glycoconjugate structures play a crucial modulatory function in regulating the host immune responses under diverse inflammatory diseases [60]. In case of COVID-19 acute respiratory infection, several recent studies evidenced the essential role of glycoconjugate structures such as heparan sulfate and sialosides (sialic acid-containing carbohydrates) in mediating the attachment and infection of SARS-CoV-2 virus [61,62,63]. This enticed the attention of researchers in devising therapeutic strategies to curtail the virus infection [41,64]. Additionally, it has also been reported that infectious SARS virions harness host histo-blood group antigens on S proteins [65], which might presumably modulate the interaction with host cell glycoprotein receptors through carbohydrate–carbohydrate interactions [66]. The current transcriptomic based research analysis is aimed to generate a hypothetical view of the alteration in glycan machinery, which could be used to explain the relevance of glycosylation underlying the lung pathology and immune responses reported in COVID-19 subjects. Given the maximum number of gene representation from the nasopharyngeal samples followed by the Calu cell line, interpretation of the DEGs relevance is largely based on the gene expression pattern observed from these samples. Results from our manual literature-based analysis reveals a common pattern of glycan epitope structures shared between SARS-CoV-2 virus and other various pathogenic viruses for adhesion and infection of the host cells. Schematic representation of these diverse glycan epitope host ligands for pathogenic virus (Appendix A, Figure 4).

Increased expression pattern of major fucosyltransferases across the nasopharyngeal and lung samples as well as the cell lines may indicate a tendency of the SARS-CoV-2 infection to augment the inflammatory response by increasing the expression of glycan epitopes such as blood group antigens and Lewis X antigens that regulate the immune cell attraction to infected tissues [67,68,69]. The gene expression pattern of sialyltransferase from the nasopharyngeal samples was also insightful into the potential immunomodulatory nature of SARS-CoV-2 virus. Gene expression data from the nasopharyngeal sample showed an interesting pattern regarding the sialyltransferases, wherein there was an increase in expression of genes coding for α-2,3 sialyltransferases and α-2,8 sialyltransferases concomitant with decreased expression of genes coding for α-2,6 sialyltransferase enzymes involved in the synthesis of sialyl-Tn antigen. From an immune perspective, these sialyltransferases are known to generate sialylated glycans involved in immune cell trafficking and inflammatory response [70,71]. Previous glycomic analysis of the human lung and respiratory tract tissues revealed predominant expression of N- and O-glycans expressing α-2,3/α-2,6/α-2,8 linked sialoglycans [72,73]. Moreover, immunomodulatory and viral adhesive functions were also attributed for these sialoglycans expressed in the human lung and respiratory tract tissues. The expression pattern of sialyltransferase reported in this study correlate with that of the metaplastic mucous cells in ferret airways infected with human H1N1 influenza virus [74].

Based on these findings we hypothesize that SARS-CoV-2 infection results in the increased expression of α-2,3 and α-2,8 linked sialic acid structures in the human respiratory tract tissues aiding viral replication as observed in other similar avian viruses [75]. Similar to the alteration in bronchial mucins reported for patients suffering from chronic bronchitis and cystic fibrosis it will be interesting to evaluate whether human airway mucins from SARS-CoV-2 infected individuals express increased levels of sLex epitope, related to inflammation and infection [76]. Increased expression of B4GALNT2 gene, coding for the core enzyme involved in the synthesis of Sd(a) antigen, across different samples infected with SARS-CoV-2 is an interesting observation. Previous studies ascertained diverse cellular roles for B4GALNT2 associated Sd(a) antigen expression on cell surfaces which includes—prevention of H1N1 viral infection, reducing metastasis of cancer cells, lytic function of cytotoxic T-lymphocytes [77]. Hence, targeted gene expression studies as well as glyco-profiling of the airway epithelium infected with SARS-CoV-2 virus can validate whether Sd(a) antigens predominate sLe^x^ antigens and the increased expression of B4GALNT2 is an adaptive mechanism to reduce the severity of the infection.

Gene expression pattern and the induced network model analysis results of the inflammatory CLECs and the anti-inflammatory SIGLECs mentioned in the current report is in alignment with the clinical phenotype of cytokine storm, neutrophil extracellular trap formation as well as the functional exhaustion of lymphocytes reported in severe cases of COVID-19 patients [46,47,48,49,78,79]. Increased gene expression pattern of inflammatory CLECs in the nasopharyngeal samples may represent the molecular markers of innate immune function of airway epithelial cells and immune components [80,81]. However, dual anti-inflammatory function have also been attributed to few CLECs such as CLEC7A (Dectin-1) and CLEC4E (Mincle) depending on the ligand and its interaction with the pattern recognition receptors [82]. Similarly, the anti-inflammatory function of differentially expressed SIGLECs to attenuate immune responses, may be an adaptive mechanism for preventing lymphocyte exhaustion or cell death under heightened inflammatory conditions [83]. Further to, a suppressive function of host antiviral innate immune response has also been attributed to few SIGLECs (such as the SIGLEC1 protein) [84], which can be hypothesized to be induced by the SARS-CoV-2 infection conditions for evading the host innate immune response. Manual compilation of previous scientific reports also enabled us to identify the functional role of few differentially regulated lectins as potential adhesion receptors for various viral strains including the SARS-CoV-2 virus. Summarized view of the differentially expressed lectins and proteoglycans along with its previously established regulatory role on innate and adaptive immune responses could aid in exploring the role of glycoconjugate structures in dysregulated airway mucosal immunity elicited by SARS-CoV-2 infection [85]. 

In the current study, results from the gene expression pattern are largely influenced by the nasopharyngeal swab samples. Since the nasopharyngeal swab samples represent a collection of diverse epithelial and immune cell types, cell type-specific analysis is essential to delineate the diverse inflammatory and anti-inflammatory functions of these glycogenes. Further analysis of the inflammatory pathways in conjunction with the experimental validation of the immune regulatory lectins should provide greater insights into the molecular mechanisms underlying compromised innate immune response in severe cases of COVID-19 patients [85]. Lectins such as CLECs and SIGLECs have been considered as potential immunotherapeutics in the field of cancer immunology and other chronic inflammatory diseases [86,87]. Small molecules generated for these lectins could be even explored for therapeutically modulating COVID-19 immune dysregulation. 

## 5. Conclusions

The current transcriptomic based study provides meaningful insights into the nature of gene expression alteration in glycosylation enzymes, proteoglycans, lectins and potential glycan markers that can be selected for targeted experiments to understand the immunomodulatory functions underlying SARS-CoV-2 infection. A list of potential CLECs, SIGLECs and Galectin lectins have been highlighted in the current research, which can be experimentally evaluated to modulate the cytokine storm or the lymphopenia pathology reported in COVID-19 subjects.

## Figures and Tables

**Figure 1 viruses-13-00082-f001:**
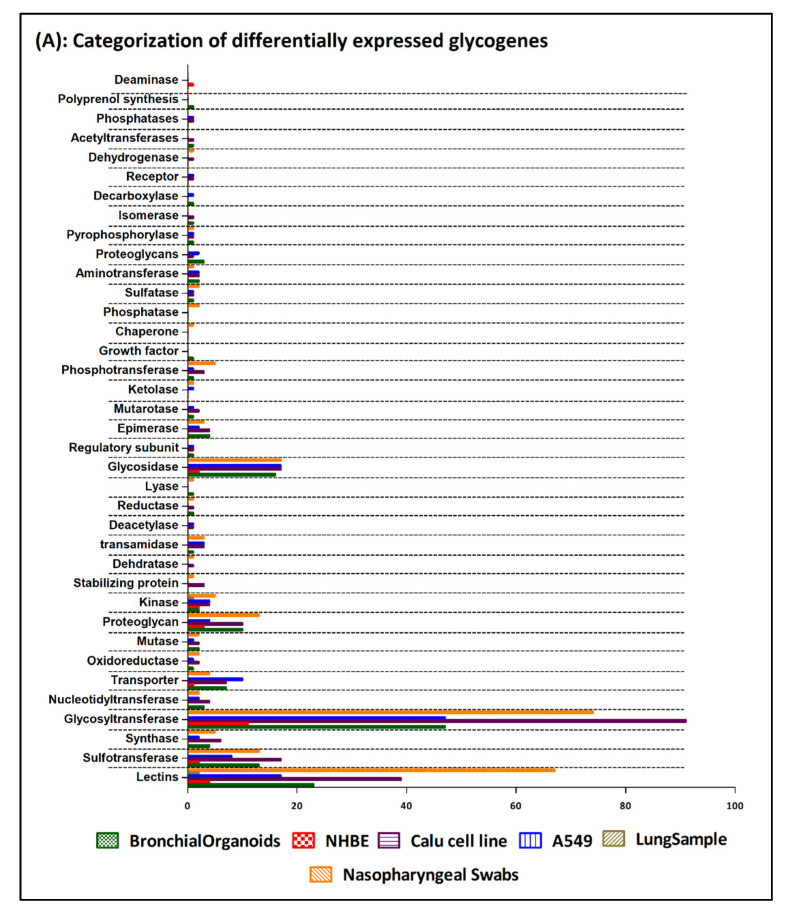
(**A**) Bar chart representing the categorization of the differentially expressed glycogenes identified from the 6 SARS-CoV-2 transcriptomic datapoints. The graph is generated using the GraphPad Prism 5 software. (**B**) Clustered heatmap generated using the Heatmap w ggplot (Galaxy version 2.2.1). Using a blue-white-red coloring scheme, clustering is performed using the default maximum similarity measure and the complete hierarchical clustering measure. The row labels represent the significantly differentially expressed glycogenes (with ≥1.5-fold change) form 6 datapoints identified from the SARS-CoV-2 infected human cell lines, organoids, ex vivo lung and nasopharyngeal samples.

**Figure 2 viruses-13-00082-f002:**
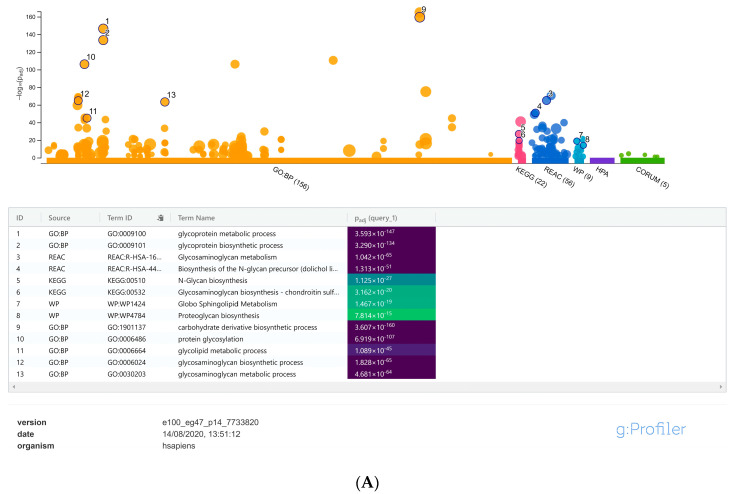
(**A**) Manhattan plot generated from the g:profiler toolset for the functional enrichment analysis using the default significant threshold measures. The x-axis shows the functional terms grouped and color-coded by the respective data sources and the corresponding enrichment *p*-values in negative log_10_ scale are illustrated on the y-axis. A more detailed result table below the image, highlights the manually selected top ranking 10 functional enriched terms and corresponding *p*-values. (**B**) More detailed result of the GO molecular function output generated from the g:Profiler tool highlighting the maximum represented categories among the input list of differentially expressed glycogenes with their corresponding enrichment *p*-values in negative log_10_ scale.

**Figure 3 viruses-13-00082-f003:**
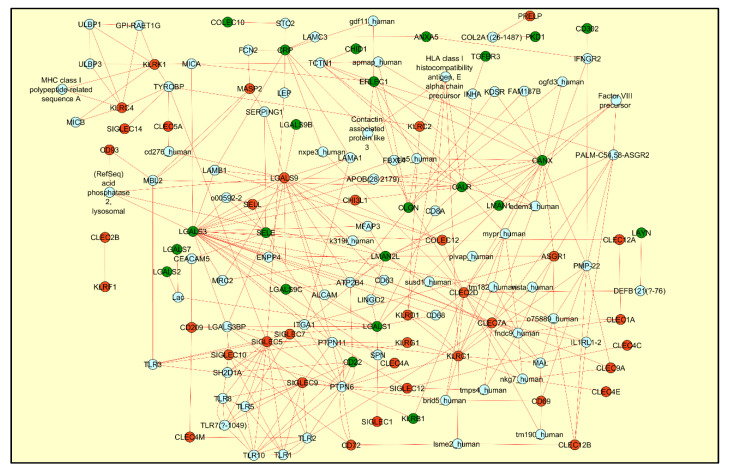
The figure represents the induced network module generated for the differentially regulated lectins using CPDB database and visualized using Cytoscape network. Transcriptomics data from the SARS-CoV-2 infected human nasopharyngeal swab sample was used to highlight the graph with red color indicating upregulated genes and the green color indicating the downregulated genes. The protein–protein interaction network, drawn in red color, reveals the association of lectins with both the positive and negative immune regulatory pathways.

**Figure 4 viruses-13-00082-f004:**
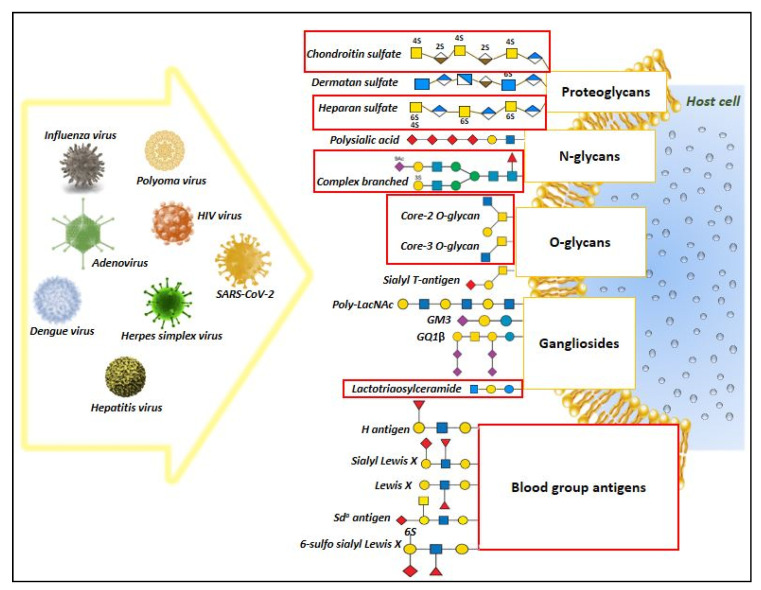
Overview of diverse glycan epitopes that act as host ligands for pathogenic virus attachment and infection. The figure represent the results generated from our manual literature search highlighting the involvement of glycan structures as host ligands for major pathogenic viruses. Among these, the red highlighted boxes represent potential glycan epitope structures that was predicted to change in response to SARS-CoV-2 infection, based on the transcriptomic data analysis, thus suggesting a possibility of shared features in glycan structures during viral infection that might aid in viral attachment and infection.

## Data Availability

Not applicable.

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
