# Peer review of "Transcriptomic Analysis of Respiratory Tissue and Cell Line Models to Examine Glycosylation Machinery during SARS-CoV-2 Infection"

_viruses, 2021, doi:10.3390/v13010082_

Round 1

Reviewer 1 Report

Oomen et al. present a bioinformatic meta analysis of transcriptomic data from patients, organoids and cell lines with respect to the changes in glycosylation (both glycans and glycoproteins) that could occur during infection by SARS-CoV-2. The authors pool differential analyses to generate a list of differentially expressed genes that are related to glycosylation, and then use a variety of gene ontology and pathway analysis tools to try to summarise the information presented.

While this may not be the fault of the authors, assessing this manuscript has been difficult because the resolution of the figures has resulted in unreadable text, and required directly consulting the supplementary tables to understand the data presented. I would recommend that the figures are re-worked so that they are clearly visible at the lower resolution, or that the production of the manuscript files preserves the resolution of the images in the paper so that the labels can be clearly read.

A: This manuscript hinges upon the identification of 298 glyco-genes as differentially expressed. All of the conclusions follow from the identification of this list, but unfortunately the description of how this list of genes was derived is lacking. As such, it is impossible to know how robust the identification of these genes is, whether these genes are likely to be truly differentially expressed, and so whether the conclusions from this manuscript are warranted. Specific problems include:

  1. 7 datasets were retrieved from GEO (according to Supp. Table 1), but it is unclear whether all 7 datasets are reflected in Supp. Table 2, which lists the fold changes in 6 different cases. Also, according to the description of the data in Supp. Table 1, there are multiple biological sources (e.g. for GSE151803, lung organoids, lung and pancreas are available) – but it is unclear which ones are used for further analysis. The description of which data is used, and how that maps to the columns of fold changes needs to be more clearly shown.
  2. There is no description of the statistical methods used to calculate what the differential genes are.
  3. It looks like a single threshold was used as a cut-off for differential expression, without taking into account the possible biological variation that may be more or less depending on the sample being tested. E.g. if you use a cut-off appropriate for cell lines, it will pick up a much larger number of differentially expressed genes, which could just as easily be attributed to natural biological variation.
  4. To understand the differential analysis, we also need to be able to place the differential genes in the context of all the other genes that are differentially regulated. E.g. for the pharyngeal swab, how many genes in total are differentially expressed? Are glyco-genes more or less differentially expressed than the other genes?
  5. It is unclear whether it is advisable to pool the differentially expressed genes from different biological backgrounds because it is unclear if any transcriptional response to infection will reflect some common behaviour of the tissues/cells being queried. At the very least, an analysis of the overlap in expression of the glyco-genes between the uninfected samples would help to understand what common glycosylation related genes have changed.
  6. An overview of the total number of glycosyltransferases, lectins and glycoproteins that are differentially expressed would be useful to see, so the reader can estimate whether the glycosylation machinery is differentially regulated, and whether the glycoproteome itself is differentially regulated.
  7. Summarising from Suppl. table 2, I estimate that around 115 glycosyltransferases are differentially expressed. This represents a little over half of the glycosyltransferases and glycan sulfotransferases in humans, and seems extraordinary. While this could be explained in part by different parts of the glyco-genome being affected in different cell systems, a discussion as to the implications of this would be appropriate.

B: In general, the enrichment and pathway analyses look to be acceptably performed, but what they reveal does not contribute very much to the overall argument of the manuscript – the enrichment does not provide any new information that is further investigated. The enrichment in Figure 2 largely reflects the selection of genes – and when nearly half of the known human glycosyltransferases are differentially expressed, wouldn’t it seem more likely this is reflecting the biases in underlying annotations in GO and Reactome.

C: The most clear examples of differential expression from the data look to be the fucosyltransferases, sialyltransferases and B4GALNT2, but any direct link to SARS-CoV-2 infection cannot be drawn (only indirectly through immunomodulation and effects of inflammation). The influence that glycosylation has upon the infection of SARS-CoV-2 is being investigated by many labs, and one of the more intriguing aspects is (e.g. the work of the Esko or Turnbull labs) the ability of the spike to bind heparin (and that heparin can block infectivity). As serglycin is differentially expressed, and it can carry heparan sulfate in mast cells, I would find a discussion of this (as well as any differential regulation of the HS pathway) to be very relevant within the manuscript.

D: Summarisation of the epitopes that have likely changed in the format of a single figure (especially the information on lines 208-224) would be tremendously helpful. This will help the reader with an understanding of the common changes that you identify as changing during infection, clearly showing features that are common to viral infection, and those that are specific to coronavirus infection.

E: I’m not sure of the value of Figures 3, 4 and 5 to the analysis in the manuscript. The figures are largely illustrative, and are not discussed in any detail. They could easily be moved to the supplementary information without disrupting the flow of the manuscript, and so can be produced in full resolution.

F: In general, the manuscript is written clearly and provides adequate background about aspects of SARS-CoV-2 infection and disease progression, and inflammation. More detail about what is currently known about the specific roles of glycosylation in this infection would assist in placing the results from the differential analysis into context.

Author Response

A: This manuscript hinges upon the identification of 298 glyco-genes as differentially expressed. All of the conclusions follow from the identification of this list, but unfortunately the description of how this list of genes was derived is lacking.

Response: Description of how the gene list is compiled is now elaborated with references in Section 2.2. Supplementary file 6 is added now detailing the total list of glycogenes used for the current analysis.

As such, it is impossible to know how robust the identification of these genes is, whether these genes are likely to be truly differentially expressed, and so whether the conclusions from this manuscript are warranted. Specific problems include:

  1. 7 datasets were retrieved from GEO (according to Supp. Table 1), but it is unclear whether all 7 datasets are reflected in Supp. Table 2, which lists the fold changes in 6 different cases. Also, according to the description of the data in Supp. Table 1, there are multiple biological sources (e.g. for GSE151803, lung organoids, lung and pancreas are available) – but it is unclear which ones are used for further analysis. The description of which data is used, and how that maps to the columns of fold changes needs to be more clearly shown.

Response 1: Supplementary Table 1 modified to reflect the data points selected for the analysis. Supplementary Table 2 header modified to reflect the samples and data points from where the data is obtained. Additional supplementary file added detailing the DESeq2 analysis results with description of the datasets used for the analysis. Further, results section is modified to bring more clarity about the data analysis results.

  1. There is no description of the statistical methods used to calculate what the differential genes are.

Response 2: Details incorporated into the results section now. The list of DEGs were generated using Genepattern software and the default Benjamini-Hochberg adjusted p-value statistical calculation was selected for the differential expression analysis.

  1. It looks like a single threshold was used as a cut-off for differential expression, without taking into account the possible biological variation that may be more or less depending on the sample being tested. E.g. if you use a cut-off appropriate for cell lines, it will pick up a much larger number of differentially expressed genes, which could just as easily be attributed to natural biological variation.

Response 3: Cut-off applied for identifying significantly differentially expressed genes was based on adj. p value ≤ 0.1. A stringent cut-off value of fold change ≤ -0.58 and ≥ 0.58 was used for focussing the analysis on glycogenes with fold change ≥ 1.5 fold. Details of the total differentially regulated genes with glycogene coverage is incorporated in the Supplementary table 4 and the results section is modified to elaborate this step.

  1. To understand the differential analysis, we also need to be able to place the differential genes in the context of all the other genes that are differentially regulated. E.g. for the pharyngeal swab, how many genes in total are differentially expressed? Are glyco-genes more or less differentially expressed than the other genes?

Response 4: Details now incorporated in the additional Supplementary table 4 with separate tables highlighting the description statistics and frequency distribution. In the results section the percentage of coverage of glycogenes with respect to other differentially expressed genes is also mentioned after modification.

  1. It is unclear whether it is advisable to pool the differentially expressed genes from different biological backgrounds because it is unclear if any transcriptional response to infection will reflect some common behaviour of the tissues/cells being queried. At the very least, an analysis of the overlap in expression of the glyco-genes between the uninfected samples would help to understand what common glycosylation related genes have changed.

Response 5: Pooling of the genes as performed was under described. Simply put, it is the grouping of genes that was performed under the cellular function association to understand the pathway representation in biological samples (biopsies and cell lines) relevant to the upper respiratory tract in response to SARS-CoV-2 infection. This is now articulated in the results section. In the Supplementary table 2, this grouping is revealed along with the overlap in expression across the samples to understand the similarity or dissimilarity in expression pattern.

  1. An overview of the total number of glycosyltransferases, lectins and glycoproteins that are differentially expressed would be useful to see, so the reader can estimate whether the glycosylation machinery is differentially regulated, and whether the glycoproteome itself is differentially regulated.

Response 6:  New graph is added in the main figure and details provided in the result section. Detailed analysis report is provided as an additional supplementary file.

  1. Summarising from Suppl. table 2, I estimate that around 115 glycosyltransferases are differentially expressed. This represents a little over half of the glycosyltransferases and glycan sulfotransferases in humans, and seems extraordinary. While this could be explained in part by different parts of the glyco-genome being affected in different cell systems, a discussion as to the implications of this would be appropriate.

Response 7: Approximately 213 genes both predicted and experimentally confirmed to encode glycosyltransferases (https://www.genenames.org/data/genegroup/#!/group/424) and 51 genes encoding sulfotransferases have been organized in HGNC database; https://www.genenames.org/data/genegroup/#!/group/821). Out of these gene list, in the current data analysis, 143 glycosyltransferases and 28 sulfotransferases were identified to be differentially expressed from 6 data points in total. With the aid of gene set enrichment analysis and gene family association, association of these genes with the glycolipid, N-glycan and O-glycan metabolic pathways were identified. This section is now little elaborated and included in the result section for clarity.

 B: In general, the enrichment and pathway analyses look to be acceptably performed, but what they reveal does not contribute very much to the overall argument of the manuscript – the enrichment does not provide any new information that is further investigated. The enrichment in Figure 2 largely reflects the selection of genes – and when nearly half of the known human glycosyltransferases are differentially expressed, wouldn’t it seem more likely this is reflecting the biases in underlying annotations in GO and Reactome.

Response B: Yes, this is one of the major limitation of the current enrichment analysis softwares. In order to overcome this limitation we also undertake the manual analysis of the possible association of DEGs with human immune system and pathogenic viruses based on available literature data.

 C: The most clear examples of differential expression from the data look to be the fucosyltransferases, sialyltransferases and B4GALNT2, but any direct link to SARS-CoV-2 infection cannot be drawn (only indirectly through immunomodulation and effects of inflammation). The influence that glycosylation has upon the infection of SARS-CoV-2 is being investigated by many labs, and one of the more intriguing aspects is (e.g. the work of the Esko or Turnbull labs) the ability of the spike to bind heparin (and that heparin can block infectivity). As serglycin is differentially expressed, and it can carry heparan sulfate in mast cells, I would find a discussion of this (as well as any differential regulation of the HS pathway) to be very relevant within the manuscript.

Response C: Yes, this is very relevant to the current observation we had. Biological role of Serglycin is an important observation as suggested by the reviewer. We have now incorporated this point into the result section.

 D: Summarisation of the epitopes that have likely changed in the format of a single figure (especially the information on lines 208-224) would be tremendously helpful. This will help the reader with an understanding of the common changes that you identify as changing during infection, clearly showing features that are common to viral infection, and those that are specific to coronavirus infection.

Response D: Additional figure is added now in the draft which provide an overview of the potential epitopes that was predicted to be affected based on gene expression data from SARS-CoV2 transcriptomic data analysis. The figure also represent the results generated from our manual literature search highlighting the involvement of these glycan structures as host ligands for other major pathogenic viruses, thus suggesting a possibility of shared features in glycan structures during viral infection that might aid in viral attachment and infection. Among these glycan epitope structures special mentioning about the possible glycan epitope structures changed specifically to SARS-CoV-2 infection is also highlighted.

E: I’m not sure of the value of Figures 3, 4 and 5 to the analysis in the manuscript. The figures are largely illustrative, and are not discussed in any detail. They could easily be moved to the supplementary information without disrupting the flow of the manuscript, and so can be produced in full resolution.

Response E: Figure 3 and 4 are now moved into the supplementary files. Figure 5 is retained as the main figure (Figure 3 now). This figure highlights the immunomodulatory links of the differentially expressed lectins and could be a major guidance network for targeted experimental validation efforts.

F: In general, the manuscript is written clearly and provides adequate background about aspects of SARS-CoV-2 infection and disease progression, and inflammation. More detail about what is currently known about the specific roles of glycosylation in this infection would assist in placing the results from the differential analysis into context

Response F: Additional references is now added in the discussion section highlighting the relevance of glycosylation in SARS-CoV-2 infection based on recent scientific evidences.

Reviewer 2 Report

Major comments.

The article by Oommen and collaborators addresses a very interesting issue in SARS-CoV-2 infections. As many other respiratory and intestinal viruses glycobiology plays a central role in viral infection at least at two different levels. Glycobiology is very relevant in the immune response against viruses and glycans are used by many viruses as receptors or co-receptors.

Oomen and collaborators have performed a transcriptomic analysis from data deposited in public repository using high-throughput functional genomics.

The results show key immunomodulatory lectins, proteoglycans and glycan epitopes related to inflammatory signaling. The authors were also able to detect putative receptors for SARS-CoV-2.

So far, this reviewer is satisfied with the work presented and do not have any negative comment against it.

Author Response

Thanking the reviewer for the positive response on our manuscript. As stated by the reviewer one of the key highlight of our manuscript is that through the focused transcriptomics analysis we were able to show key immunomodulatory lectins, proteoglycans and glycan epitopes related to inflammatory signalling in response to SARS-CoV-2 infection from tissues and cell lines relevant to upper respiratory tract. 

Round 2

Reviewer 1 Report

The authors have added details to their methods section describing how the differential gene analysis was performed. While I still have reservations about the choice of cut-off (and the subsequent number of genes that are detected as differentially regulated), the choice of cut-off is clearly presented in the text, and a reader can fairly interpret the results. In particular, Supplemental Table 2 is very useful in understanding the results.

The extra references and context help to place these results into context within the broader studies of glycosylation and SARS-CoV-2.